# A Multimodal Platform for Simultaneous T-cell Imaging, Defined Activation, and Mechanobiological Characterization

**DOI:** 10.3390/cells10020235

**Published:** 2021-01-25

**Authors:** Martin Fölser, Viktoria Motsch, René Platzer, Johannes B. Huppa, Gerhard J. Schütz

**Affiliations:** 1Institute of Applied Physics, TU Wien, 1060 Vienna, Austria; foelser@iap.tuwien.ac.at (M.F.); viktoria.motsch@boku.ac.at (V.M.); 2Institute of Agricultural Engineering, University of Natural Resources and Life Sciences, 1190 Vienna, Austria; 3Institute for Hygiene and Applied Immunology, Center for Pathophysiology, Infectiology and Immunology, Medical University of Vienna, 1090 Vienna, Austria; rene.platzer@meduniwien.ac.at (R.P.); johannes.huppa@meduniwien.ac.at (J.B.H.)

**Keywords:** T-cell, immunological synapse, atomic force microscopy, total internal fluorescence microscopy, calcium imaging

## Abstract

T-cell antigen recognition is accompanied by extensive morphological rearrangements of the contact zone between the T-cell and the antigen-presenting cell (APC). This process involves binding of the T-cell receptor (TCR) complex to antigenic peptides presented via MHC on the APC surface, the interaction of costimulatory and adhesion proteins, remodeling of the actin cytoskeleton, and the initiation of downstream signaling processes such as the release of intracellular calcium. However, multiparametric time-resolved analysis of these processes is hampered by the difficulty in recording the different readout modalities at high quality in parallel. In this study, we present a platform for simultaneous quantification of TCR distribution via total internal reflection fluorescence microscopy, of intracellular calcium levels, and of T-cell-exerted forces via atomic force microscopy (AFM). In our method, AFM cantilevers were used to bring single T-cells into contact with the activating surface. We designed the platform specifically to enable the study of T-cell triggering via functionalized fluid-supported lipid bilayers, which represent a widely accepted model system to stimulate T-cells in an antigen-specific manner. In this paper, we showcase the possibilities of this platform using primary transgenic T-cells triggered specifically via their cognate antigen presented by MHCII.

## 1. Introduction

T-cells recognize antigen presented at the surface of antigen-presenting cells (APCs) with exquisite sensitivity and specificity. At the onset of this process, the T-cell receptor (TCR) complex binds with the rather low-affinity peptide-loaded major histocompatibility complex (pMHC). While the mechanisms for transducing TCR-pMHC binding into a subcellular signal are still enigmatic [1,2], this process eventually leads to the phosphorylation of immunoreceptor tyrosine-based activation motifs (ITAMs) on the TCR-associated CD3 subunits. A variety of non-exclusive models have been proposed to explain signal transduction, including the kinetic proofreading of single TCR-pMHC binding events [3,4], the serial triggering of multiple TCR complexes by single pMHC molecules [5], or the kinetic segregation of bulky proteins from the narrow cleft of the immunological synapse [6]. Recently, data have also indicated mechanical force sensor capabilities in the TCR complex [7]. Yet, to this date, the extent to which the different models contribute to the correct picture of T-cell activation is not clear.

The still-enigmatic nature of TCR signal transduction is at least attributable in part to experimental challenges. First, a large proportion of experimental data has been recorded using antibodies instead of the natural ligand pMHC for triggering T-cell activation. Because antibodies engage TCRs with considerably higher affinity and via epitopes that differ considerably from those of pMHC, the mechanisms underlying activation likely differ from those of a physiological stimulus. Second, early T-cell activation proceeds within a few tens of seconds upon first contact between cells and the activating surface, and is hence experimentally difficult to capture. Third, different phases in the T-cell activation process can be observed, ranging from timescales of minutes [8] to hours [9], which reflects the predominance of different molecular mechanisms. Fourth, it is difficult to disentangle contributions from mechanisms acting simultaneously, such as integrin-mediated adhesion, costimuli, and the actual antigen recognition event.

To account for these aspects of T-cell antigen recognition, we have developed an experimental platform for antigen-specific T-cell activation as a means to monitor the very first stages of T-cell triggering at high spatiotemporal resolution. To this end, we employed non-activating tipless AFM cantilevers as a nano-actuators for controlled touchdown of single T-cells to functionalized surfaces. In this fashion, our methodology provides precise control over the time-point of first contact. In addition, it allows for a multiparametric assessment of the activation process by the quantification of forces exerted by the T-cell, the readout of intracellular calcium levels as a second messenger, and the high-resolution imaging of the immunological synapse using total internal reflection fluorescence (TIRF) microscopy. We showcase the method of using glass-supported fluid lipid bilayers as mimicry of antigen-presenting cells [10], which carry adjustable concentrations of pMHC, adhesion molecules, and costimulatory molecules.

## 2. Materials and Methods

### 2.1. Ethical Compliance Statement

All animal experimentation (related to breeding, sacrifice for T-cell isolation) was evaluated by the ethics committee of the Medical University of Vienna and approved by the Federal Ministry of Science, Research and Economy, BMWFW (BMWFW-66.009/0378-WF/V/3b/2016). Animal husbandry and experimentation were performed under national laws (Federal Ministry of Science, Research and Economy, Vienna, Austria) and the ethics committee of the Medical University of Vienna, and according to the guidelines of the Federation of Laboratory Animal Science Associations (FELASA).

### 2.2. T-cell Isolation and Transduction

Primary T-cells were obtained as previously described [11]. Briefly, T-cells were isolated from lymph nodes of 5c.c7 TCR transgenic mice and stimulated with 2 µM HPLC-purified moth cytochrome C (MCC) peptide (ANERADLIAYLKQATK, Intavis Bioanalytical Instruments) in full medium (RPMI 1640 (Gibco) containing 10% FCS (Sigma), 100 U/mL penicillin/streptomycin (Gibco), 2mM glutamate (Gibco), 1 mM sodium pyruvate (Gibco), 1x non-essential amino acids (Gibco), and 50 µM β-Mercaptoenthanol (Gibco)). On day 2, culture volume was doubled and 100 U/mL IL-2 (eBioscience) was added. On days 3 and 5, T-cell cultures were expanded in a ratio of 1:1. Dead cells were removed by centrifugation trough a Histopaque-1119 (Sigma) cushion on day 6. T-cell experiments were conducted on days 7–9 after initial stimulation.

### 2.3. Protein Expression and Functionalization

I-E^k^ and MCC peptide were prepared and labeled as described previously [11,12]. H57-scFv was expressed in inclusion bodies, refolded, site-specifically labeled with maleimide-functionalized AF647 (Thermo Fisher Scientific, Waltham, MA, USA), and purified as described [11]. All experiments were performed on I-E^k^ loaded with AF555-conjugated MCC.

### 2.4. Preparation of Sample Plates

For our experiments, we used No. 1.5 glass cover slips (Menzel) that had been plasma-cleaned (PDC-002, Harrick Plasma) for 10 min. Frames for the sample chambers were produced by pouring addition-curing duplicating silicone (twinsil soft, Picodent) into a home-made 3D-printed mold (Appendix A; see Appendix A for CAD drawing). After hardening, frames were removed from the mold. We applied a thin layer of addition-curing duplicating silicone to each frame and attached them with light pressure to the cover-slips. After 5 min of hardening, the measurement chambers were ready to use.

Supported lipid bilayers (SLB) were prepared as described previously [11]. In short, 93 µg of 1-Palmitoyl-2-oleoyl-snglycero-3-phosphocholine (POPC, Avanti) and 2.6 µg of 1,2-dioleoyl-snglycero-3-[N(5-amino-1-carboxypentyl)iminodiacetic acid] succinyl[nickel salt] (Avanti) were dissolved in chloroform and mixed. After evaporation of the chloroform under a nitrogen stream, vesicles were formed by resuspending the lipid mixture in 1 mL 1× PBS (Sigma) and bath-sonicating for 15 min (USC500TH, VWR). The solution was kept at 4 °C for up to 2 weeks.

Prior to the experiment, the vesicle solution was incubated in the measurement chamber for 20 min and then rinsed extensively with PBS. Bilayers were incubated for 75 min with His_12_-I-E^k^/MCC, His_10_-B7-1 (optional, 32 ng/mL, Sinobiological) and His_10_-ICAM-1 (optional, 25 ng/mL, Sinobiological) diluted in PBS. Before each experiment, we tested each bilayer for fluorescence recovery after photobleaching of the fluorescently labelled I-E^k^/MCC. Only SLBs with mobile fractions >80% were taken for the experiments.

I-E^k^/MCC concentrations of 15 ng/mL and 0.75 ng/mL were chosen to achieve I-E^k^/MCC surface densities of approximately 100 molecules/µm^2^ (denoted as high I-E^k^/MCC) and of approximately 5 I-E^k^/MCC/µm^2^ (denoted as low I-E^k^/MCC). I-E^k^/MCC densities were determined by dividing the bulk fluorescence signal by the single-molecule fluorescence, as described previously [13]. Briefly, single-molecule fluorescence signals were determined from images of SLBs containing very low densities of fluorescently labelled I-E^k^/MCC, so that well-separated single-molecule signals could be identified. Single-molecule positions were determined using the Crocker-Grier algorithm [14] We summed up all pixel values in the raw data that were within in a 7pixel-diameter foreground circle around each signal’s position. Local background was estimated by calculating the mean camera count per pixel in a ring around the signal circle (excluding any pixels affected by nearby signals). The background was subtracted pixelwise both from the single-molecule foreground signal and the bulk signal.

For surface PEGylation of the storage chamber, we incubated it with poly-L-lysine(20)-g[3.5]-PEG(2) (Susos) for 75 min. Both chambers were extensively washed before use.

For measuring T-cells directly seeded to SLBs, we used 8-well chambers (Nunc Lab-Tek) instead of the home-built chambers. Here the wells were removed from the original glass slides and attached to plasma-cleaned glass cover slips using addition-curing duplicating silicone (twinsil soft, Picodent).

### 2.5. Optical Microscopy and AFM

Drawings of the beam paths are shown in Appendix A. All images were recorded using a Zeiss Axiovert 200 inverted microscope equipped with a JPK Cellhesion 200 and a back-illuminated EM-CCD camera (Andor iXon DU897). Samples were illuminated and imaged through a 100× oil-immersion objective (NA = 1.46, Zeiss Plan-Apochromat), yielding a pixel size of 160 nm in the object plane. For TCR imaging, we coupled a LAS 638 nm laser (Oxxius) in TIRF mode into the objective. The calcium indicator FURA-2 was illuminated subsequently with two LEDs (340 nm, 375 nm, Thorlabs) overlaid and coupled into the pathway of the laser using dichroic beam splitters (DM1: F38-376, DM2: F38-488, Semrock, respectively); FURA-2 imaging was performed in epi-configuration.

For our experiments, we used a dichroic mirror (DM3: zt488/640rpc, Chroma) and an emission filter (EM1: ZET488/640m-TRF, Chroma). Emitted light was split into two pathways (Cairn OptoSplit II; D4: FF64-FDi01 Semrock; EM2 FF01-525/45-25, Semrock; EM3: ET675/50m, Chroma). A lens (f = −100 mm, PCV, Edmund optics) was installed in the Fura-2 emission pathway to shift the focal plane by 4.5 µm into the center of the T-cell.

AFM cantilevers (µMasch HQ:CSC38 / tipless / no coating / dagger-shaped / nominal force constants: 0.09 N/m, 0.03 N/m, 0.05 N/m) were cleaned in chloroform overnight and plasma-cleaned in residual air for at least 30 min. Cantilevers were submerged in poly-D-lysine (70–150 kDa, Sigma Aldrich) overnight. Each cantilever was used for pick-up and transfer of a single cell only; cantilevers could be reused multiple times after cleaning.

### 2.6. Multiparametric Imaging of T-cell Activation

T-cells were prepared for experiments using imaging buffer (Hank’s buffered salt solution (Sigma) +2% fetal bovine serum (S181H, Biowest)) on ice. Cells were washed, labelled simultaneously with Fura-2-AM (5 µg/mL, Molecular Probes) and H57 antibody (5 µg/mL) for 20 min, and washed twice. The buffer in the measurement and storage chamber was exchanged for imaging buffer, and T-cells were seeded to the storage chamber immediately after washing. We waited until no floating cells could be seen before starting experiments (typically 10 min). Cells were kept in the storage chamber for up to 120 min. All measurements were performed at room temperature.

The coated cantilever was installed on the AFM (JPK Cellhesion 200) and calibrated using JPK’s contactless calibration procedure. In all experiments, the AFM acquisition frequency was 10,000 Hz. Using cell-capture mode, the cantilever was lowered toward the T-cell in steps of 1 µm at a speed of 5 µm/s; seizing of Brownian motion was taken as indication of contact with the cantilever. We chose T-cells for pickup that showed the characteristic round shape, and avoided cells with obvious signs of apoptosis.

We recorded images according to the following illumination protocol: in one illumination cycle, samples were illuminated for 2.5 s with the 340 nm LED (calcium imaging), for 1 ms with the red laser (TCR imaging), for 150 ms with the 375 nm LED (calcium imaging), and for 1 ms with the red laser (TCR imaging). This cycle was repeated every 4 s, covering a total observation period of approx. 10 min.

### 2.7. Data Analysis

All data was analyzed with custom python code (version 3.6) utilizing the following libraries: numpy, scipy, pandas, opencv-python, matplotlib and seaborn [15,16]. The code is available upon request from the corresponding author.

For calcium analysis, we subtracted the background from images taken at 340 nm and 375 nm, and calculated the pixel-wise ratio. We took as the calcium signal the mean of all pixels within a distance of 3.2 µm from the center of the cell. This value was taken conservatively to ensure that the analyzed region covered only parts within the T-cells.

To analyze the TCR signal, we first averaged the two images taken per illumination cycle. Image noise was reduced by applying Gaussian blur with a kernel size of five pixels. Cell-contact areas were determined by applying a user-defined threshold to the TCR signal frame.

Characteristic transition times for activation, deflection, and area are defined throughout the paper as the time-point of half maximal effect. For this, we determined the maximum value, $f\left(t_{2}\right)$, and the minimum value, $f\left(t_{1}\right),$ with $t_{1}<t_{2}$, and calculated the time point $t_{median}$ at the half amplitude $f\left(t_{median}\right)=\left(f\left(t_{1}\right)+f\left(t_{2}\right)\right)/2$. In case of ambiguities, we chose the time-point closer to $t_{2}$ for $t_{median}$. Particularly, this procedure turned out to be stable with respect to the choice of $t_{1}$ and $t_{2}$.

To synchronize different traces, we shifted the time coordinate, with t = 0 being the time-point of activation. The time-point of first contact was determined manually by the appearance of a TCR signal in the TIRF image; this ensured the capture of the first part of the adhesion events, when T-cells contact the SLB via the tips of individual microvilli.

Offset and drift of AFM deflection data was compensated by fitting a spline to data obtained prior to interaction between T-cell and bilayer and subtracting the spline from the original data. To reduce noise, all shown AFM data represent averages obtained with a window size of 1 s.

Calcium signals were normalized by dividing every individual trace by its mean basal value. Figure 2b shows the median calcium signals of individual synchronized time traces. The shaded area corresponds to the 95% confidence interval obtained by bootstrapping using 1000 subsamples.

Average areas in Figure 3a are the medians of individual synchronized time traces. The shaded area corresponds to the 95% confidence intervals.

Comparisons of distributions were computed using the two-sided Kolmogorov–Smirnov test on two samples (scipy.stats.ks_2samp).

## 3. Results

All experiments were performed on primary, antigen-experienced murine CD4^+^ effector T-cells transgenic for the 5c.c7 TCR, which specifically recognize moth cytochrome C presented in the context of the MHC class II protein I-E^k^ (I-E^k^/MCC)[11]. Prior to experiments, T-cells were loaded with the ratiometric calcium indicator Fura-2 and labeled with a monovalent single-chain antibody fragment derived from the TCRβ-specific mAb H57-597 site-specifically conjugated to Alexa Fluor 647 (H57-AF647). H57-AF647 labels the TCR without interfering with I-E^k^/MCC binding [11].

Our approach was based on a two-chamber plate (Figure 1a), which we devised to integrate the following demands: The storage chamber should be sufficiently inert to keep T-cells non-activated for up to two hours, whereas the measurement chamber should allow for antigen-specific T-cell activation upon lowering individual cells towards the functionalized surface. Prior to an experiment, each plate was built from a silicon frame (Appendix A) that had been glued onto a glass coverslip to enable simultaneous fluorescence microscopy (from the bottom) and AFM experiments (from the top). The barrier between the two chambers needed to be sufficiently high to allow for separation of the two chambers without spillover during sample preparation, and sufficiently low to enable its flooding before transferring the cells from the storage to the measurement chamber; here we used barriers with 1 mm height and 2 mm width. The overall height of the frame was 3 mm. The storage chamber was PEGylated to facilitate the easy pick-up of single T-cells via the AFM cantilever, and at the same time to prevent T-cell pre-activation prior to the actual experiment. The measurement chamber featured a fluid-supported lipid bilayer functionalized with I-E^k^/MCC, and optionally the adhesion molecule ICAM-1 and the co-stimulatory molecule B7-1.

In a typical workflow, we first seeded T-cells into the storage chamber, and selected a single cell using transmission light microscopy. A tipless poly-D-lysine-coated AFM cantilever was lowered toward the selected T-cell until contact was established, and the T-cell was allowed to adhere to the AFM tip for 3 min. At the time-point of first contact, we kept the force exerted onto the selected T-cell as small as possible (<0.3 nN). We verified via calcium imaging whether the contact between the T-cell and the cantilever itself led to activation; on average, only ~7% of the cells showed increased calcium levels after pickup. These cells were excluded from further analysis. Next, we transferred the adhered T-cell via the cantilever from the storage chamber to the measurement chamber, and lowered the T-cell toward the functionalized supported lipid bilayer. Cell lifting and transfer to the measurement chamber took a few seconds. The approach to the SLB was done with incrementally smaller step sizes, lasting in total from 40 to 450 s. During and after the approach, we recorded and monitored three characteristic signals: the cantilever deflection (reporting pushing and pulling forces), the calcium signal, and the fluorescence signal originated from H57-AF647 decorated TCRs. As described in the Materials and Methods section, we used epi-fluorescence excitation to record the calcium signal, and TIRF excitation to record the TCR signal; biplane imaging allowed for focusing simultaneously on the lipid bilayer (TCR channel) and the cell center (calcium channel). As soon as a signal appeared in the TCR channel, indicating the contact of the T-cell with the supported lipid bilayer, we stopped the approach and halted the AFM cantilever at a constant height; in particular, we did not push the T-cell toward the SLB. A typical example of a single T-cell brought into contact with a strongly activating supported lipid bilayer is shown in Figure 1c and Video S1. After a phase of cellular tip-toeing [17]—the formation of transient small-scale contacts via individual microvilli characteristic for scanning T-cells [18,19]—we observed a sudden increase in the contact area, accompanied by an increase in the calcium signal. At the same time, we recorded the cantilever deflection during the adhesion and activation process.

Based on the three characteristic signals, we accepted 40% of the cells for further analysis. In particular, we did not analyze situations in which two or more cells or non-cell material had adhered simultaneously to the cantilever (11%), in which the cell unintentionally prematurely contacted the SLB (3%), in which cells were obviously not alive (6%), in which cells displayed an elevated Fura-2 ratio prior to contact with the bilayer (7%), in which cells showed excessive movement (2%), in which we did not achieve a sufficient signal to noise ratio (10%), and in which cells did not show a calcium flux during the contact time (21%).

First, we investigated whether T-cells stimulated via the multimodal platform showed a similar calcium response compared to T-cells activated conventionally by seeding them onto functionalized SLBs. For this, we used strongly activating SLBs functionalized with a high density of I-E^k^/MCC, ICAM-1, and B7-1. On average, the calcium response was initiated with similar kinetics and reached similar magnitudes, irrespective of whether cells were delivered via the AFM cantilever or cells were seeded directly onto the activating surface (Figure 2). A slightly smaller spread in the time between first contact and calcium flux for cells seeded directly to the surface was noted, which may indicate higher flexibility in the contact formation. Also, the formation of TCR microclusters was readily observable in TIRF microscopy (Appendix A), indicating that neither the pick-up process nor contact enforcement affected early T-cell activation behavior.

Cell spreading on activating surfaces is considered as a second hallmark for early T-cell signaling [18,20,21]. We investigated the formation of close contacts between the T-cell surface and functionalized SLBs using TIRF microscopy of the TCR specifically labeled using H57-AF647. Immediately upon contacting the surface, T-cells were observed to form individual, small, well-isolated, and highly dynamic contacts with the SLB, as had been described previously [18]. After a few minutes, cells responded with a sudden increase in contact area (Figure 3a). For quantitative comparison, we define here the time-point of an increase in calcium or contact area as the time-point of the half-maximal effect (see Materials and Methods), yielding concurrence of contact formation and calcium signal (Figure 3b). For better visualization of the correlation in the kinetics between calcium signal and contact area, we plotted the two values in a parametric plot, with time being the parameter (Figure 3c and Appendix A). Interestingly, while there were instances of low contact area at high Fura-2 ratios, we did not observe the opposite, i.e., high contact area at low Fura-2 ratio, in line with elevated calcium levels preceding [22] or even representing a precondition for T-cell adhesion [23].

Delivery of single T-cells via the AFM cantilever allowed us to probe the interaction of T-cells with SLBs that were less potent for activation. Omitting ICAM-1 reduced the average area per cell after adhesion (Figure 3a), without any observable effect on the magnitude and kinetics of the calcium signal (Figure 2a and Figure 3b). We only observed an approximately 50% reduction of the number of activated cells (Appendix A). However, reducing the stimulus further by omitting both ICAM-1 and B7-1 and reducing the I-E^k^/MCC surface density to 5 molecules per µm^2^ yielded strong effects on the response. First, and in line with a previous report [24], the calcium response was delayed (Figure 2a) and lower in magnitude (Figure 2b). Second, the contact area was considerably reduced in size, with only a few dynamic contact points visible, likely reflecting the tips of microvilli contacting the SLB. In this case, no clear increase in the contact area was observable; while some cells responded by the formation of a lamellipodium, others did not show any detectable alteration of the contact phenotype (Appendix A).

We then determined the force exerted by T-cells force onto the cantilever. According to Newton’s third axiom (i.e., *actio est reactio*) [25], such force directly corresponds to the total normal force exerted via a T-cell onto the SLB in the opposite direction. When confronting T-cells with highly stimulatory SLBs (high levels of I-E^k^/MCC as well as ICAM-1 and B7-1), we observed predominantly pulling forces with a magnitude in the single-digit nN range (Figure 4, first row; see Appendix A for exemplary traces). The onset of force exertion coincided with the time-point of cell spreading and the initiation of a calcium signal. While it is plausible thaT-cells can pull perpendicularly on SLBs decorated with ICAM-1, such pulling is less likely for SLBs coated with I-E^k^/MCC and B7-1 only. Indeed, in such cases, T-cells were observed to also exert pushing forces (Figure 4, second row), which started before cell adhesion and the calcium signal.

Further lowering the SLB’s stimulatory potency by employing I-E^k^/MCC at low densities and omitting both ICAM-1 and B7-1 led to virtually complete abrogation of forces exerted onto the SLB (Figure 4, third row). Under such conditions, T-cells applied only minute pulls and pushes in the range of a few hundred pN. Of note, minute pushes exerted by the cells corresponded directly to transient area increases, lasting over several minutes (Figure 5). Such behavior appears reminiscent of the tip-toeing phase of T-cells first encountering an activating surface via microvilli before cytosolic calcium release.

## 4. Discussion

We have described a method for temporally defined stimulation of T-cells via functionalized SLBs, using an AFM cantilever as a nanoactuator for controlled T-cell delivery and manipulation. Using this method, T-cell activation proceeds similarly compared to T-cells seeded on SLBs. After some tens of seconds of probing the surface, a sudden rise of intracellular calcium levels can be observed that is accompanied by the spreading of the cell and the formation of TCR microclusters.

Compared to conventional seeding of T-cells onto SLBs, however, this method offers a number of critical advantages:It allows for parallel readout not only of commonly recorded signals such as the TCR distribution or intracellular calcium levels, but also of pushing and pulling forces exerted by the T-cells on the SLB. A number of previous reports have highlighted the relevance of force in T-cell antigen recognition [26,27,28,29,30,31,32,33]. In agreement with existing literature, we observed T-cell-imposed forces amounting to magnitudes of up to 3 nN upon the T-cell making contact with stimulating SLBs [33].High-resolution imaging experiments can be extended in duration when T-cells fail to establish sufficient adhesive interactions for spreading on opposing surfaces. When monitored with the use of conventional experimental protocols, T-cells typically migrate quickly out of the field of view, thereby complicating long-term observation. Due to the arrest on the AFM cantilever in our platform, even single-molecule tracking experiments are feasible in TIRF configuration without the need for stable cell adhesion and spreading.T-cells can be brought into contact with an SLB at a precise user-defined time-point. The appearance of the fluorescence TCR signal within the TIR-based excitation field can be used as a direct indication of the T-cell making contact with the SLB.Within the crowded yet highly dynamic environment of the lymph nodes’ T-cell zones, T-cell migration to antigen-presenting cells (APCs) is guided by the fibroblastic reticular cell network [34]. Our experimental platform allows the emulation of such situations by pushing the T-cell towards the SLB or retracting it away from the surface, thereby offering access to new observation parameters.

Taken together, we consider our novel platform to be an interesting complementation of the methodological toolbox for studying early T-cell activation, which offers the possibility to contextualize force information with synapse morphology and the T-cell activation state.

## Figures and Tables

**Figure 1 cells-10-00235-f001:**
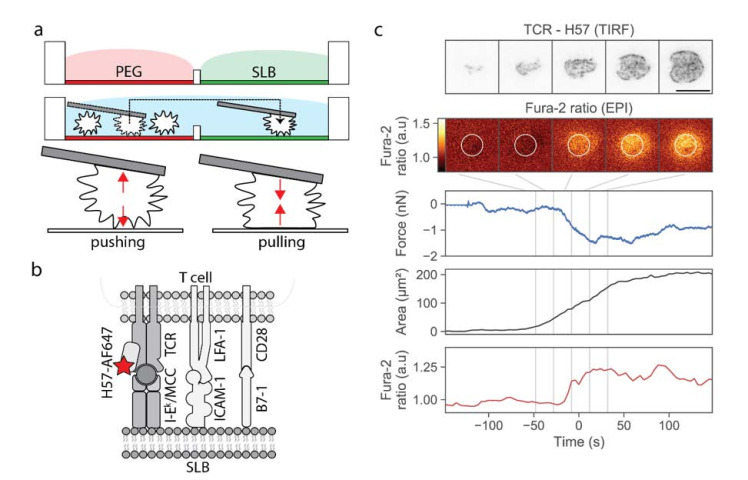
Design of the multimodal platform for simultaneous T-cell imaging, defined activation, and mechanobiological characterization. (**a**) A two-chamber platform was used to create two distinct and separated surfaces, an inert PEGylated surface and a functionalized SLB, which can be connected by flooding the barrier. Single T-cells were selected from the inert surface, picked up via a poly-D-lysine-coated AFM cantilever, and transferred to the SLB-coated chamber. (**b**) SLBs were functionalized with I-E^k^/MCC, the adhesion molecule ICAM-1, and the costimulatory molecule B7-1. The TCR was labeled with H57-Alexa 647 (red star). (**c**) Exemplary data obtained from the approach of one T-cell to an activating SLB functionalized with I-E^k^/MCC, ICAM-1 and B7-1. We show five characteristic images of the TCR-channel in TIRF configuration and the Fura-2 ratio in epi-configuration, recorded at the indicated time-points. In the calcium image, we indicated the analysis region as a white circle. In addition, the deflection, the quantified single-cell contact area, and the Fura-2 ratio is shown for the whole approach curve. Time-point zero is defined via the half-maximum value of the calcium signal. Scalebar: 10 µm.

**Figure 2 cells-10-00235-f002:**
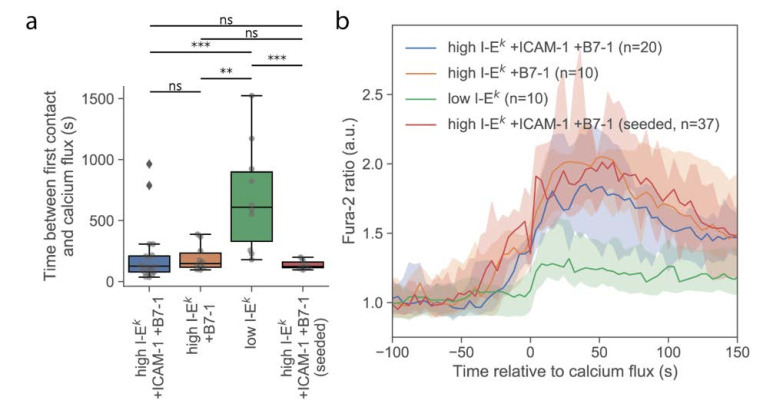
Analysis of the calcium signals. (**a**) Duration of the T-cell contact preceding a rise in intracellular calcium. Data are shown as Whisker box plots indicating the interquartile range (box), median (line), and the individual data points corresponding to single cells (circles). (**, *p* < 10^−2^; ***, *p* < 10^−3^) (n = 20, 10, 10, and 7 cells from left to right). (**b**) Median Fura-2 ratio of activating T-cells for given conditions. Shaded regions were determined via bootstrapping.

**Figure 3 cells-10-00235-f003:**
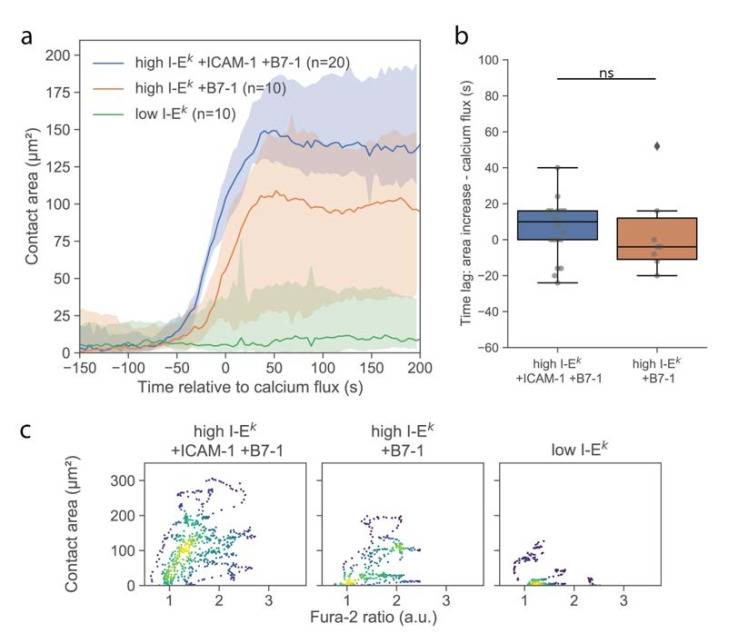
Analysis of the T-cell contact area. (**a**) Median contact area as function of time. Before averaging, the time axis of each data set was shifted by the half-maximal time of the calcium signal. Shaded areas represent 95% confidence intervals. (**b**) Lag between half maximal area increase and calcium signal. Data are shown as Whisker box plots indicating the interquartile range (box), median (line), and the individual data points corresponding to single cell (circles) for high I-E^k^/MCC + ICAM-1 + B7-1 (blue, n = 20 cells) and high I-E^k^/MCC + B7-1 (orange, n = 10 cells). (**c**) Contact area of individual T-cells was plotted against the corresponding Fura-2 signal in a parametric plot, where time is the parameter. For each cell, we included 37 data points recorded in the time interval, ranging from 50 s before to 100 s after the calcium increase. Colors indicate density of data points. Data are shown for T-cells interacting with SLBs featuring ICAM-1, B7-1 and high I-E^k^/MCC (n = 20 cells), B7-1 and high I-E^k^/MCC (n = 10 cells), and low I-E^k^/MCC (n = 10 cells) (ns, *p* > 0.05).

**Figure 4 cells-10-00235-f004:**
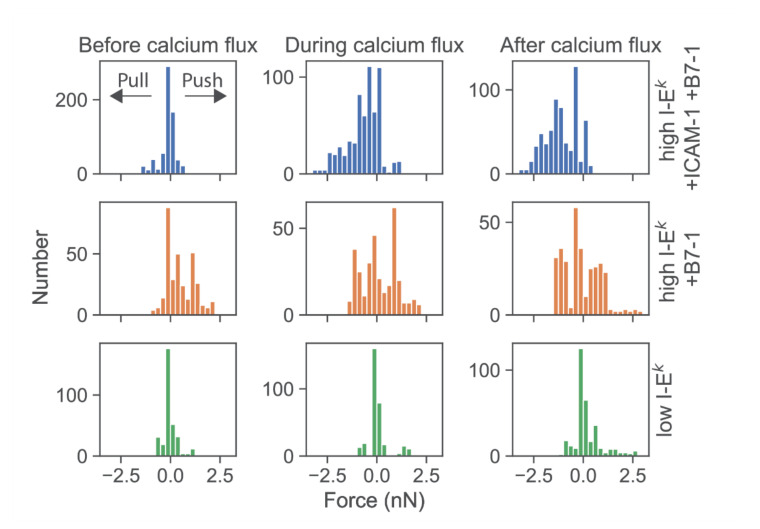
Deflections of the AFM cantilever due to pushing and pulling of single T-cells. Histograms of all force values were recorded from single T-cells from t = −50 s to t = 50 s. Prior to analysis, time axes were shifted relative to the half-maximal value of the calcium signal. Data were pooled for time intervals before calcium flux (−50 s ≤ t ≤ −16 s), during calcium flux (−16 s < t ≤ +16 s), and after calcium flux (16 s < t ≤ 50 s). Data are shown for T-cells contacting SLBs featuring ICAM-1, B7-1 and high levels of I-E^k^/MCC (blue, n = 20 cells), B7-1 and high levels of I-E^k^/MCC (orange, n = 10 cells), or low levels of I-E^k^/MCC (green, n = 10 cells).

**Figure 5 cells-10-00235-f005:**
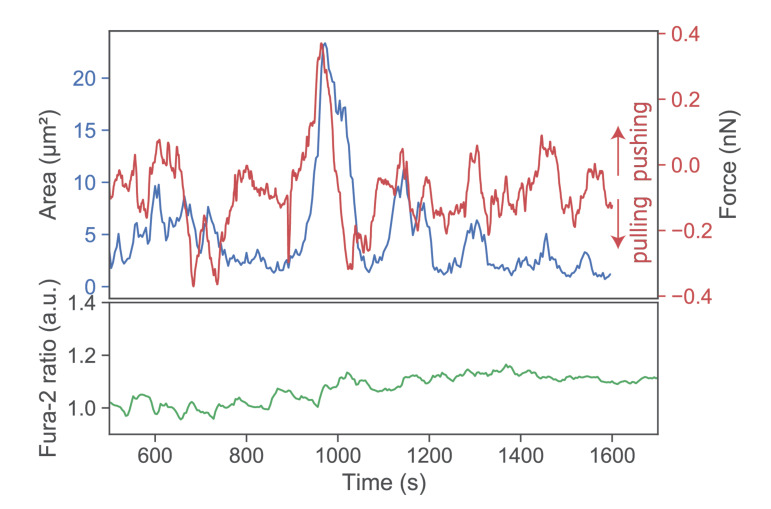
Exemplary pushes and pulls by a single T-cell. A T-cell was lowered toward an SLB functionalized with low I-Ek/MCC only. Contact area (blue), force signal (red), and Fura-2 ratio (green) are shown for a representative time period >10 min.

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
