# Peer review of "A Multimodal Platform for Simultaneous T-cell Imaging, Defined Activation, and Mechanobiological Characterization"

_cells, 2021, doi:10.3390/cells10020235_

Round 1

Reviewer 1 Report

Report on : A multimodal platform for simultaneous T cell imaging, defined activation, and mechanobiological characterization by Martin Fölser, Viktoria Motsch, René Platzer, Johannes B. Huppa, Gerhard J. Schütz

This article presents a combination of AFM, epifluorescence and TIRFM to study the early mechantransduction of T cells on model SLB decorated with suitable MHC molecules. It is a very interesting and refined methodology with a large potential, and the data presented is coherent with the current understanding of this highly complex moment of early recongition by a competent immune cell of an antigen presenting cell.

Nevertheless, I have several questions and comments that I would like to see adressed and discussed before publication, and if possible, I am willing to re-read the modified manuscript upon re-submission.

I will refer to the page and line numbers of the pdf I received from the review system for clarity.

p2, l59 : The definition of the zero time for contact is a complex point of view since it depends on the way it is evaluated. T cells, in particular primary ones, are covered with villis, which may collapse at very low forces. This is a limit for the detection of early contacts under forces that is often overlook and I will come back on that point later on (see below).

p2, l69-75 - note : this paragraph should be aligned.

p2, l85 : No resting times for the cells after the centrifugation is mentioned. This should be added if there is one in the present protocole. Also temperatures should be precised. Also the state of the cells (resting, primed) should be precised since theyr are at first stimulated with MCC peptide.

p2, l86-... : Precisions on the mold printing and motive should be given since the present paper presents a new methodology. I would then push for having more details in the M&M section about the set-up, in particular on the sample preparation side. Dimensions of the mold (size of the wells, hights of the barriers) should be given for the reader, if experimentalist, to realize how easy / hard is the proposed procedure. This could take the form of a figure where sketches and photos of the wells could be detailled. A missing step in the description and which appears to me the more difficult is the manipulation and glueing of the frames on the coverslide. Moreover, the measuring chamber is not described.

p3, l91-96 : How is the fluidity and continuity of the SLB assessed ? The fluidity is something that is talked about in the manuscript and which is quite important here. From experience, I know that it is rather difficult to have something very reproducible.

p3, l102-103 : Please detail how are measured the densities of molecules on the SLBs and comment on the reproducibility of the procedure.

p3, l108 : Please precise "silicone" methodology (see above my comment on the measuring chamber)

p3, p115 : For the description of the setup, I personnally think that a schematics of it, with the placement of the very clever lens for epi would be a plus. Using the text alone, it took me some time to verify what I infered from first reading, that is that Ca++ measurments are made in epi... This should be clarified in the text. For TIRFM, the depth of penetration that is achieved / choosen should be given.

p3, l128-130 : Shape and spring constant of the levers should be given. Plasma type (residual air, oxygen or else) and oven should be precised. "D" of polylysine should be explained (why not "L") and this should be consistent through the MS (I have noticed at least one "L" version in the text - p5, l204). Coating type of the lever should be precised since It may affect the reflectivity and drift of the spring, but also the coupling of lever bending with epi / TIRF (see Cazaux et al. https://pubmed.ncbi.nlm.nih.gov/26521163/).

p3, l134 : Is it Fura-2-AM ? This should be precised. As noted before, no resting time after dye loading or details of the procedure itself is given. This should be corrected, IMHO. also, the homogenity of the dye loading should be presented and discussed (eg. histogram of initial cell fluorescence)

p4, l140 : "Sizing of Brownian motion" is the indication of contact. Does that mean that capture is made at almost zero force or what is the typical force for capture ? Of course, the usual 1nN at least that people use in SCFS is too large for the T cell and may even activate it non specifically. Note : the text says "smooth texture" and cells are shown spiky on the schematics : the text could be modified not to perturb non immuno-friendly readers. For AFM measurements, acquisition frequencies, temperature, speed of lever motion are missing...

p4, l143 : I am wondering if the first step of 2.5sec is (or not) a typo considereing the much shorter times of illumination in the following steps. No comment is made neither on the power of the epi light that is used nor on possible coupling effects with lever bending if the power is high (since the coating info is missing on the lever presentation, see comment above).

p4, l148 : If the code is open / available on Git, this should be mentionned, python version should be stated, plus the main librairies that are used.

p4, l151 : the 3.2µm / center distance cutoff should be explained : why is it not an average on the whole cell projected area ? What does this brings to the quantification ? To the S/N of the measurements or their robustness ? Since this article presents a new technique, such details should be given, IMHO.

p4, l161 : Since TIRF images appear to be grainy and a bit noisy, is there any image processing involved in the alignment to automatise it (using eg. averaging and thresholding) or is it done, due to the small number of cell traces involved here, by eye ?

p4, l165 : Why is there an average over 1 sec ? Aside, the acquisition frequency is not precised...

p4, l167 vs p4, l170 : Is there a reason to use once the median and once the mean ? What is the package used for bootstrapping ?

p5, l183 : The interaction site of the labelling antibody should be precised for the reader to understand that it is suffisently far away from all binding sites on the TCR chain and that it does not introduce any geometric hindrance in order to be a faithful reporter of all TCR positions.

p5, l194 : Did one try to perform the same pushing experiments on PLL substrate directly after capture as a control ? They could be informative about really non specific effects. Aside, what is the duration of the transfert of cell from one well to the other, and what is the total min / max delay between capture and measure (again, this is a resting time that may be crucial for the reproducibility of the experiment, and may be one of the cause of the 17% of non varying fluo cells for Ca++).

A question that comes directly from the previous comment is : and what happens on a second pressing of the same cell ? can one peel it from the substrate and press it again ? Does the cell stay on the substrate on retraction ? What are the levels of forces at separation for the different cases, which would be in my mind heavily influenced by the presence of integrin ligands ? All this data exists in the force vs time signals and could be processed and presented in supp mat for completion.

Second question : can a cantilever be reused for capturing and pressing several cells in a sample, how and what are the controls made for it ot be reused ? This is a question that will interst many experimentalist, not only in terms of stats but also in terms of costs...

p, fig1 : what is not really clear when reading the manuscript is what is the amount of force to establidh the contact during SCFS. When reading Fig1c, it could be infered to be zero force, but the precision has to be made in the text. I will also add, for the sake of clarity, the mode of imaging with the molecule (TIRFM for TCR, epi for Fura2). The calcium color coding has no scale, which has to be corrected. In the caption, it is witten PLL and not PDL (see above) and the mode of imaging for calcium is not clear (again)... please correct. Maybe, adding a circle of the 3.2µm radius in the epi images may help to understand better the quantification here (because one could have simply used the full square image, for example, of a given size, larger that the cell itself, and presented the mean intensity over the square, without having to struggle for centering proper

p6, l225 : "As soon as a signal appears in the TCR channel" - please refer to my revious comment on p4, l161. Here, more details have to be included for the reader to understand the process.

p6, l228-231 : When summed, it appears that only 81% (53+5+6+17) of the cells are discussed. What are the last 20% ? Can the author comment on the fact that 1/2 of the cells only where included in the study ? Working with immune cells, I totally agree that it is often difficult to have more that 75% of cells doing something clear and comparable, in particular in term of Ca++ fluxes at single cell level, in particular when one knows the inhomogeneity of the dye loading process, see my comments above - Did the authors use the basal fluo level of the cells too to select them for further experiment or only the morphology as mentionned in the text ?

p6, l232 : Could the authors define tip-toeing here ? I still think that villis will vanish when the cell will spread, so a good definition of tip-toeing would help the reader to picture the phenomena at play (in ref 15, there is no force except gravity). Is tiptoeing detected as TRIF signal fluctuations (maybe the fluo images may contain more info that presented here, even with 4sec time interval...)

p6, fig2 : On part a, Transparents points (on all figures presenting boxplots) would make it easier to see their dispersion. When comparing with part b, the numbers of cells included in the data are different : can the authors comment (and correct if needed) on why is it so (and for some cases very largely : 7 vs 37 for the seede cells). One thing should be added in the caption : one point is one cell. The slopes (on the average curves) for calcium raise appear to be similar, only the max ratio seems to vary. May be a comment should be made about that ? an intersting point is that from start to max, the rise appears to last one minute, which is consitent with litterature (and my own experiments, I thank you for this !)

p7, l244 : It would be informative to have typical force vs time curves for the different cases, in suppl mat.

p7, l249 : the authors are mentionning the appearence of clusters but no data is presented. This is a very important point, worth showing the images and their quantification eg. sizes, lifetimes, motion and so (again, the question of threshold will be raised here). I would advocate for the authors to add this data to strengthen the manuscript and help this very interesting technique to be more convincing for the general audience.

p7, fig3 : On a, I find that the "time relative to calcium flux" to be weakly defined in the text, and this should be corrected. Here, quantification of the slope of area vs time, which appear to be different on the mean curves could be presented. Again, time lag of b is ill defined, and from the presented data is it difficult ot me to assess the ***. Could the author comment on that ? I may have missed some details. On c, is it a point a cell (in that case i don't understand, since n values provided in the caption are rather low) or a dot-like heatmap (in that case a classical Seaborn lib map in Python could be better, provided that a color scale with units is provided, which is not the case here). On c, a correlation could also be presented using correlation plots or Pearson like tests (again, available in Seaborn / Python - https://seaborn.pydata.org/examples/many_pairwise_correlations.html).

p8, l266 : "Immediately... [16]" : movies and analysis should be shown to support this assertion that, to me, is too vague (see my comments in the same line above).

p8, l270 : Correlation not being causality, this part should be rewritten / discussed / amended

p8, l278 : The authors could present the fraction of activation / calcium flux positive cells depending on the surface to support te statements here.

p8, l280 : "delayed" - this should be supported by data, in my opinion, citing a figure or presenting numbers in the main text.

p8, l 284 : I am not convinced by the supplementary fig1 which should be better detailled in its caption or in the text. The color coding of the plots is explained. Two type of spreading morphologies are, from my point of view, seen on these panels : large continuous spreading vs punctuate contacts... May be the fraction of each type of even could be shown in the figure to support the proposition present in the main text ?

Here, there is, to my mind, a missing experimental case that the authors could add to complement their finding nicely, namely High I-Ek alone. This could allow to state the dose dependance effects and the combination of molecules ones.

p8, fig4 : On a, the differences of shapes of the histograms are very intersting to me (and the previous missing condition could bring a lot here) but are not commented or discussed in the main text. Changing the molecules changes the patterns of forces over the first 3 min encompassing the calcium fluxes : could this anaylisis be refined by presenting the histograms of forces before, during and after calcium rise as a function of the molecule(s) present on the substrate ? The last lines ('Data...') of the caption are not claera and should be rewritten, to my mind.

I would have added the time traces, with zooms if needed, that are present in suppl. fig 2 here, before the histograms for the readers to see the heterogeneity of the behaviours : if a cell exhibit very large positive forces, then it may stay in that zone and never go back to zero or negative. The pooling of the data presented in a is difficult, to me : you pool over different cells, but also over time, over activation,... maybe shwing an histogram for a typical cell would be more simple and easier to figure out what is happening here, and disentangle it from pooling effects that may affect the shape of the total histogram. Also, on a, the label on y axis is not descriptive.

p9, fig5 : One could also presented the area / time correlation signal direclty, and also correlation with calcium signal which varies only by 20%, which is not a lot (but is often seen in experiments, due to non homogeneous loading of the cells)

p10, l322 : TCR microclusters are never shown here... (see my above comments : remove it or add data)

p10, l337 : The precision of the contact time depends on the way the contact is defined when looking at the details (see my previous questions). This should be modified accordingly to your responses / additions to my previous comments

p10, l345-346 : This very vague sentence should be rewritten...

Supplementary fig 2 is difficult to read and should be remade.

Supplementary movie is surprisingly not playing in VLC...

Reviewer 2 Report

Although I felt that there were some things that could be clarified and some errors to be rectified I enjoyed this manuscript and feel the method is useful and could provide novel insights in the future. In particular I felt the low I-Ek data was hard to compare with the data with higher I-Ek because costimulus and adhesion molecules were also present in this case.

General comments:

  • The start and end point of spreading and calcium are manually determined. Although the authors say this was stable with respect to choice of start and end this is quite vague and unconvincing. Ideally these should have been determined algorithmically or at the very least the criteria for selecting the start and end should have been stated. If they could be stated they could be encoded to be performed automatically, which would aid both with objectivity and in making analysis less labour intensive.
  • Fig 2a seems to show that the seeded cells have a much tighter distribution of contact time to calcium flux. Is this the case and if so why is this? Could it be because you may stop the cells on the cantilever too far away from the surface for the cells to effectively spread? Or that holding them at a specific distance creates a force for them to work against when they are spreading?
  • Low I-Ek is not a good comparison to the other data sets since two key parameters: pMHC density and costimulus (or costimulus + adhesion molecules) are varied. This makes it difficult to draw conclusions from differences and it would be much better to compare low I-EK with B7-1 or with B7-1 and ICAM-1.
  • Neither Supplementary Fig 2 nor the supplementary video were referenced in the main text. There should be some reference to them if they are to be included.

Specific comments

  • Line 53: "Mediated" rather than "Medicated"
  • Line 231: Should cells that do not show a calcium signal be included since they represent non-activated cells, which may be useful to quantify for each different condition.
  • Lines 231-232: Should this be Fig 1c rather than Fig 1b (which is a diagram of the activation surface)
  • Fig 3b: why are the low I-Ek not plotted on this graph? What do the asterices indicate here? There should be an indication in the figure legend. Also since you used the K-S measure rather than a T test some may be confused as to why there is significance and that you specifically state thre is no difference in magnitude on lines 276-277. This is likely due to the difference in the distribution width (which the K-S also tests but the standard T test does not) rather than the mean. It would be useful to make some mention of this in the test to avoid confusion.
  • Fig 3c: There are much more points that the n number indicated in the legend. What do the data points actually correspond to? Also the colours indicated in the legend do not correspond to the colours in the plot. It seems a 'viridis' colour map was used but there is no indication of what the colour map indicates or what the scale of the colour map is, etc.
    Lines 271-273: Although Fig 3c shows a lack of cells with high contact area and low Fura-2 ratio, there were positive lag times (shown in Fig 3b) between cell spreading and calcium flux, which would not suggest calcium flux was a precondition for cell spreading as the authors state. This may be due to the half-maximal calcium flux and spreading times being used in Fig 3b rather than onset times. Whether this is the case or not, the contradiction should be addressed.
